# Developing Radio-Frequency Roasting Protocols for Almonds Based on Quality Evaluations

**DOI:** 10.3390/foods11131885

**Published:** 2022-06-25

**Authors:** Ting-Yu Lian, Su-Der Chen

**Affiliations:** Department of Food Science, National Ilan University, Number 1, Section 1, Shen-Lung Road, Yilan 26041, Taiwan; j6j61215@gmail.com

**Keywords:** almonds, radio frequency (RF), roasting, aroma, quality

## Abstract

Hot air roasting is a popular method for preparing almonds, but it takes a long time. We roasted almonds via dielectric heating using 5 kW, 40.68 MHz batch radio-frequency (RF) equipment and analyzed their quality and aroma using a gas chromatography/ion mobility spectrometer and sensory evaluation. Almonds with an initial moisture content of 8.47% (w.b.) were heated at an RF electrode gap of 10 cm; the target roasting temperature of 120 °C was achieved at weights of 0.5, 1, 1.5, and 2 kg for 4, 3.5, 7.5, and 11 min, respectively; and the moisture content was reduced to less than 2% (w.b.). For comparison, 1 kg of almonds was roasted in a 105 °C conventional oven for 120 min. The darker color and lower moisture content, water activity, and acid value of the RF-roasted almonds were favorable for preservation. The aroma analysis using gas chromatography/ion mobility spectroscopy (GC–IMS) revealed that the aroma signal after roasting was richer than that of raw almonds, and principal component analysis (PCA) demonstrated that the aromas of roasted and commercial almonds were similar. The RF-roasted almonds presented a better flavor, texture, and overall preferability compared to commercial almonds. RF heating could be used in the food industry to roast nuts.

## 1. Introduction

Almonds have a high nutritional value and are high in unsaturated fatty acids and vitamins. They are consumed after being roasted, a process which increases their hardness and crispness, enhances their aroma, and gives them a roasted appearance. Roasting can also deactivate enzymes, eliminate pests, and kill pathogenic microbes [1]. The most common type of oven is a traditional hot air oven, and the roasting method comprises hot air heating at over 100 °C for 120 min, which is extremely time-consuming. This method may lack a pasteurization effect and requires a lengthy processing time, increasing the processing cost. Furthermore, prolonged roasting can easily produce an undesirable flavor [1]. Therefore, new processing strategies must be developed to achieve an increased roasting speed while maintaining superior product quality.

Yang et al. [2] used infrared radiation (IR) combined with hot air technology to roast almonds. Ten grams of almonds were heated to the final roasting temperature of 130 °C by IR within 1 min and were then roasted with hot air for 15 min. The initial roasting time was reduced by more than 80%; the bacterial reduction increased by 38%; and the sensory quality was not significantly changed (appearance, texture, taste, and overall acceptability). Agila et al. [3] roasted 50 g of almonds at 177 °C for 5 min before roasting them in a microwave oven at 135 °C for 5 min (the final temperature of the almonds was 108 °C). According to the results and sensory evaluations, the aroma of the microwave-roasted almonds was superior to that of the oven-roasted almonds. Kosoko et al. [4] roasted 2.5 kg of cashew kernels in a hot air and halogen oven at 200 °C for 40 min. The rapid temperature cycle of the halogen oven effectively reduced the moisture content of the nuts. The halogen-oven-roasted cashews were highly acceptable according to the entire sensory evaluation. However, infrared and microwave heating penetration depths are minimal, and the thickness and quantity of the treated samples are limited. In the two above mentioned tests, for example, the weight of roasted almonds was only 10 g and 50 g, which is insufficient for large-scale production. Although the heating effect of a halogen oven is superior to that of a standard hot air oven for industrial applications, the short heating distance prevents it from producing a significant heating effect and thus from effectively reducing the processing time [5].

In recent years, radio-frequency (RF) heating has been used to roast nuts such as almonds [1], peanuts [6], and cashews [5,7]. RF roasting is a fast dielectric heating method that operates at frequencies ranging from 10 MHz to 300 MHz. The electromagnetic field is rapidly transformed by the top and lower electrode plates, causing the polar molecules and charged ions in the sample to violently rotate and shake, resulting in frictional heat generation. Furthermore, due to the deep penetration of RF waves, the heating is fast and uniform, reducing the heating time and improving product quality [8]. RF treatment has great potential as a new nut-roasting method because it can achieve disinfestation, pasteurization, and drying effects when the target temperature of 120 °C is reached.

The objectives of this study were to develop an RF roasting protocol; compare the quality of almonds roasted using commercial, oven, and RF heating methods; and evaluate the aroma and sensory quality of RF-roasted almonds. 

## 2. Materials and Methods

### 2.1. Materials

The almonds used in this study were purchased from Beans Group Foods Science and Technology Co. (Taoyuan, Taiwan), and their origin was California, USA. 1,1-Diphenyl-2-picryl hydrazyl (DPPH), ascorbic acid (vitamin C), synthetic glacial acetic acid, ethylenediaminetetraacetic acid (EDTA), and antioxidant butylated hydroxyl anisole (BHA) were purchased from Sigma Chemical Co. (St. Louis, MO, USA). Ethanol (95%), diethyl ether, phenolphthalein indicator, potassium hydroxide (KOH), and potassium hydrogen phthalate (KHP) were purchased from WAKO Pure Chemical Industries, Ltd. (Osaka, Japan).

### 2.2. Roasting Methods

#### 2.2.1. Determination of RF Roasting Conditions

The RF system with hot air equipment (Yh-Da Biotech Co., LTD., Yilan, Taiwan) (Figure 1) was designed to produce 5 kW and operate at 40.68 MHz, with a hot air temperature of 100 °C. The hot air was blown in from the right side. Samples of 0.5 and 1 kg were placed in a polypropylene (PP) plastic basket with holes 23 cm in diameter and 8 cm in height, and 1.5 and 2 kg specimens were placed in a larger PP plastic basket with holes 27 cm in diameter and 9.5 cm in height. RF treatment was performed with different electrode plate gaps. Because the RF equipment had a maximum current of 1.6 Amp and a maximum output power of 5 kW, the average output power of the RF system was calculated by reading the current (A) and using the following formula: power output (kW) = (5/1.6) A.

The surface temperature of the sample was measured at the center and 5 cm on either side of the center by a multifunctional infrared thermometer (Testo104-IR, Hot Instruments Co., LTD., New Taipei City, Taiwan), and the final surface temperature of the sample was also measured by an infrared thermometer (TIM-03, HILA International Inc., Taipei, Taiwan). To obtain the temperature profile, the increase in the temperature for different electrode plate gaps was recorded every 30 s until the surface temperature of the sample reached 120 °C. The dry-basis moisture content change was measured according to the weight change in the sample during the RF roasting.

#### 2.2.2. Conventional-Oven-Roasted Almonds

The 1 kg sample was placed in a single layer on a stainless-steel tray and roasted in an oven (Channel DCM-45, Taiwan) at 105 °C for 120 min; the temperature and weight of the sample were recorded.

### 2.3. Analytical Methods

#### 2.3.1. Moisture Content

We weighed 5 g of ground almonds in an aluminum dish using an electronic precision scale (HDW-15L, Hengxin Metrology Technology Co., Ltd., Yilan, Taiwan), dried them in an oven at 105 °C for 12 h, and then removed and weighed them after reaching a constant weight.

#### 2.3.2. Water Activity

The water activity (Aw) was measured using a water-activity analyzer (HC2-AW, Rotronic Instruments Ltd., Zwillikon, Switzerland). The ground almond samples were loaded into the analyzer and measured at 25 °C for 10 min each time, and the data were recorded.

#### 2.3.3. Color Measurement

The color of the samples was measured with a color difference meter (Hunter LAB, Color Flex, Virginia, VA, USA) and standardized against a white calibration plate (X = 82.48, Y = 84.23, Z = 99.61, L* = 92.93, a* = −1.26, b* = 1.17). The parameters determined were the degree of lightness (L*), redness (+a*), greenness (−a*), yellowness (+b*), and blueness (−b*). All experiments were performed in six repetitions.

#### 2.3.4. DPPH Radical Scavenging Ability Assays

We extracted 2 mL of the supernatant using a focused ultrasonic machine (20 k Hz, 1400 W, Ultrasonic Co., Ltd., New Taipei City, Taiwan) and subjected it to a DPPH assay according to [9]. We applied 20 mg/mL ascorbic acid, BHA, and EDTA as standards. All experiments were performed in three repetitions.

#### 2.3.5. Acid Value

The acid value was determined according to the Chinese National Standard analytical methods for edible oil (CNS 3647 N6082) [10]. All experiments were performed in four repetitions. 

#### 2.3.6. GC–IMS Analysis

The aroma analysis by GC–IMS was based on the method of Thomas et al. [11]. One gram of each sample was placed in a 20 mL headspace vial. The sample was heated in a heater at 50 °C for 20 min by the autosampler system, which was equipped with a 1 mL syringe with an injection rate of 170 μL/s. The sample was then injected into the gas chromatography/ion mobility spectroscopy machine (GC–IMS, Flavour Spec^®^, GAS Dortmund, Germany) in the non-diversion mode using a 20 m long 0.53 nm ID non-polar capillary column (CC), model OV-5 (5%-diphenyl, 95%-dimethylpolysiloxane), operated at 45 °C. The volatile gases were separated by injecting them into the GC using nitrogen (purity ≥ 99.999%) as the carrier gas. The gradient of the nitrogen flow rate after injection was as follows: 0~5 min—increased from 2 mL/min to 15 mL/min; 5~5.5 min—maintained at 15 mL/min, 5.5~13 min—increased from 15 mL/min to 30 mL/min; and 13~20 min—decreased from 30 mL/min to 2 mL/min. Subsequently, the gas was separated into the IMS ionization zone chamber. Data analysis was carried out using IMS Control TFTP Server software provided by G.A.S (Dortmund, Germany). 

#### 2.3.7. Sensory Evaluation

A panel of 65 people conducted a nine-point hedonic sensory evaluation of the roasted almonds in regard to appearance (how much they liked the appearance of the samples in terms of size, thickness, and completeness); aroma (how much they liked the strong smell of the samples); flavor (how much they liked the taste of the samples in terms of sourness, sweetness, and bitterness); taste (how much they liked the texture, chewiness, and crispness of the samples); aftertaste (the aftertaste after swallowing); and overall performance. All samples were allocated a three-digit number and selected randomly, and a nine-point scale for sensory evaluation was applied, with a score of 1 indicating very much dislike, 5 indicating neither like nor dislike, and 9 indicating very much like.

### 2.4. Statistical Analysis

Experimental results are expressed as means ± standard deviation (SD). One-way analysis of variance (ANOVA) was performed and subsequently subjected to Duncan’s multiple range tests of treatment mean using Statistical Analysis System (SAS 9.4, SAS Institute, Cary, NC, USA), and the significance level was set at 0.05.

## 3. Results and Discussion

### 3.1. Effect of Almond Loading on RF Heating

The effect of the RF electrode plate gap on the output power for almonds at varying loading capacities is illustrated in Figure 2. The RF energy output for 1, 1.5, and 2 kg almonds decreased as the RF electrode plate gap increased, similar to the results of Chen et al. [12], who used 1, 2, and 3 kg rice bran at an electrode plate gap of 6–16 cm, with the smaller gap demonstrating a higher RF output power for heavier rice bran. The same pattern was observed in our study. The RF output power was high for different loading levels of almonds at a lower electrode plate gap of 10 cm; therefore, a 10 cm gap was chosen to compare the RF roasting conditions for different loading levels of almonds.

The drying and heating curves of the almonds loaded at different amounts with a 10 cm RF electrode plate gap (Figure 3) showed that for 0.5 and 1 kg almonds, the temperature increased to 100 °C within 2 min of RF heating, before the rate of heating slowed down significantly until reaching 120 °C, but the moisture content of the almonds decreased more rapidly. 

For 1.5 and 2 kg almonds, a rapid increase in temperature was observed at the beginning of roasting, followed by a steady increase. However, due to the increased loading of almonds, the RF roasting time also increased. The slowest heating rate for 2 kg almonds required only 10 min to reach 120 °C, and 1 kg almonds demonstrated the fastest heating rate of 33.37 °C/min, requiring only 3.5 min to reach 120 °C and achieve the roasting effect.

Figure 4 shows the heating and drying curves of the almonds roasted in a conventional hot air oven. The temperature of the almonds was affected by heat transfer resistance in the later stages. After the almond temperature rose to 80 °C at 40 min, it took twice as long to increase the temperature by less than 20 °C, and the maximum temperature of the almonds reached only 95 °C at the end of the baking. Although the dry-basis moisture content of the almonds showed a steady decreasing trend, the dry-basis moisture content of the final almond sample only decreased to 0.043 kg water/kg dry material, and it took nearly 40 times longer than the RF technique. The drying efficiency was poor, and the high moisture content made the almonds less crisp.

The moisture content of the RF-roasted almonds did not drop significantly due to the low temperature at the beginning, but a higher moisture content could provide more polar water molecules to quickly raise the temperature. At the beginning of RF heating, the almond temperature rose rapidly in a straight line, and in the later stages most of the energy was turned into moisture evaporation heat, which was accompanied by a lower moisture content and also caused the temperature rise to slow down. Therefore, the final RF-roasted almond temperature reached about 120 °C, and the moisture content was reduced to less than 2%, which was far from the initial moisture content of 8.47% in the dried almonds before roasting. This result is in agreement with that of Wang et al. [13], who observed that the moisture content of hazelnuts in shells with RF heating decreased significantly from 34% to 19% in the initial stage (1 min) and slowed down to reach 10% in the later stages (after 2 min), while it took 22 h to reach approximately 10% if the hazelnuts were dried only by hot air at 40 °C. During the later stages of the conventional roasting of the samples, as the moisture content was already very low, causing heat conduction resistance, the thicker samples in particular required a very long roasting and drying time, whereas the RF roasting of almonds demonstrated more efficient heating.

### 3.2. Processing Performance of RF Roasting 

The electrode plate gap was negatively correlated with the RF output power, but the sample loading amount was positively correlated with the RF output power (Figure 2). However, considering the heating efficiency and uniformity of the sample, 1 kg of almonds was chosen as the loading amount to achieve the fastest roasting conditions and heating treatment with three different RF electrode plate gaps (9, 10, and 11 cm) (Figure 5). The temperature of the almonds increased linearly as the gap between the RF electrode plates increased. The almonds were heated faster at a gap of 9 cm between the electrode plates, but the heating uniformity was poor. The heating speed was slower at a gap of 11 cm between the electrode plates. An electrode plate gap of 10 cm was chosen as the operating conditions for the subsequent RF roasting of almonds.

The results were similar to those of Jiao et al. [6], who determined the appropriate electrode gap for peanuts by considering the heating uniformity: the peanuts reached 90 °C after 5.5, 11, and 18 min at gaps of 9, 10, and 11 cm, respectively. The smaller the RF electrode plate gap, the faster the heating, but considering the heating efficiency and uniformity, the electrode plate gap of 10 cm was selected as the subsequent operating conditions. Figure 3 also shows the drying curve of these three RF heating gaps: the dry-basis moisture content of the almonds at gaps of 9, 10, and 11 cm decreased from 0.092 kg water/kg dry material to 0.017, 0.026, and 0.035 kg water/kg dry material, respectively. The moisture evaporated rapidly, and the loss of moisture raised the hardness and brittleness of the almonds. The results of the texture profile analysis (TPA) conducted by Xu et al. [1] showed that after roasting, the moisture content of almonds decreased, while the hardness increased. 

In addition, the surface temperature of the almonds was measured using an infrared thermometer (Figure 6). The average temperature for the three gaps was higher than the target temperature of 120 °C, and the temperature distribution was very uniform. This is also a characteristic of RF heating and roasting—because the sample heats up quickly and the heat distribution is very uniform, it is less likely to contain cold spots.

### 3.3. Quality, Aroma, and Sensory Evaluation of RF-Roasted Almonds 

The roasting process increases the crispness of nuts, gives them a roasted color, and produces a special roasted flavor. Table 1 showed that the moisture content and water activity of the almonds were significantly reduced after roasting. The moisture content (w.b.) of the almonds roasted using hot-air-assisted RF heating for 3.5 min decreased from 8.47% to 1.57%, and the water activity decreased from 0.74 to 0.34. The moisture content and water activity results were lower than those of commercial roasted almonds and almonds roasted in an oven for 120 min, which already had a moisture content below 5.80% for the safe storage of nuts [4]. While almonds with a water activity of 0.2 to 0.3 have a longer shelf life [1], less water is more favorable for nut storage. Oil quality changes are a very important indicator of nut shelf life. The acid value tended to decrease after roasting, with the lowest acid value of 0.34 mg KOH/g obtained for RF-roasted almonds, probably due to the rapidity of RF heating, which inactivated lipase and reduced free fatty acid production.

In terms of color (Table 1), roasting reduced the reddish and yellowish color of the almonds, and the brightness was lowest in the oven-roasted almonds. Although the RF roasting temperature of 120 °C was higher than the oven roasting temperature of 105 °C, the color change in the almonds was closer to that of the untreated almonds, due to the short duration of the RF roasting.

In terms of the ability to scavenge DPPH free radicals, the antioxidant capacity of the untreated almonds was the highest, due to the fact that the roasting process breaks down the cells, followed by hot air roasting. RF roasting had a higher final temperature, and so the DPPH antioxidant effect was poorer, but it was better compared to the commercial roasted almonds. The results were similar to those of Liao et al. [5], who roasted cashew kernels with a thickness of 5 cm using RF (120–130 °C, 30 min) and hot air (140 °C, 33 min), determining that raw cashews had the best antioxidant capacity, while there was no significant difference between the RF and hot air treatments (*p* > 0.05).

Figure 7 shows the GC–IMS analysis of the volatile organic compounds in raw, RF-roasted, oven-roasted, and commercial roasted almonds. The red area indicates more volatile components, and the darker the color, the more components, while the blue area indicates the opposite. As shown in Figure 7, most of the signals appeared at the retention time of 0~150 s and the drift time of 1.0–1.5. The raw almonds showed fewer and weaker odor signals, while the roasted almonds clearly produced more odor signals, especially those roasted at a higher RF temperature (120 °C), which could be clearly seen at the retention time of 0–100 s and the drift time of 1.0–1.5. This was probably due to the fact that the RF roasting method provided heat to the almonds both internally and externally, and the almonds received heat from more sources, thus producing a higher volatile content, which conducted heat mainly on the surface of the almonds. This resulted in differences in aroma presentation, and it has been found that microwaves generate richer volatile compounds than ovens and frying [3]. 

According to the differences in the volatile compound signals in the GC–IMS results, the fingerprint profiles of 32 characteristic compounds were further selected (Figure 8). The main volatile compounds of the raw almonds were found in the 1–15 and 28–32 fingerprint profiles. Most of the roasted almonds retained their original flavor, while in 16–26, the flavor signal had the highest intensity for the almonds roasted at an RF temperature of 120 °C. In 27–32, 100 °C oven roasting and commercially available 100 °C hot air roast produced strong flavor signals, indicating that almonds need to be roasted at high temperatures to produce a special aroma.

The principal component analysis (PCA) in Figure 9 shows that the aroma of the almonds before and after RF roasting was different, while oven roasting and commercially available hot air roasting were both conducted at 100 °C and with slow heat transfer, so the aroma was similar. The difference between the aroma of the commercially available almonds and that of the RF almonds was probably due to the fact that the almonds were roasted quickly by RF at a temperature of 120 °C.

The results were similar to those of Liao et al. [7], who investigated the changes in the aroma content of cashew nuts during roasting and found that the aroma concentration increased with time and with higher temperatures. This may be related to the formation of volatiles in roasted nuts due to the Mena reaction, which cause aroma production and changes in flavor. Xu et al. [1] also analyzed the aroma of raw and hot-air-assisted-RF-roasted almonds using GC–MS and found that they contained 61 and 87 volatile components, respectively, and that the concentration of the roasted flavor components such as aldehydes, ketones, esters, alcohols, furans, pyrroles, and pyridine derivatives increased significantly after roasting, with most of the new volatile compounds commonly considered hot roasted flavors in oily nuts. Moreover, the sensory characteristics, nutritional quality, and oxidative stability of roasted macadamia nuts were greatly improved compared to raw nuts during storage [14].

Table 2 shows the 9-point-scale hedonic sensory evaluation by 65 tasters. Comparing the results of the commercial and RF-roasted almonds, there was no significant difference in appearance, aroma, or aftertaste; however, the flavor, texture, and overall acceptance of the RF-roasted almonds scored significantly higher than those of the commercial almonds, and the flavor score of 6.63 was significantly higher than the commercial almonds’ 6.03 points. This may be due to the above principal component analysis, which demonstrated that the RF-roasted almonds produced several special aromas that were different to those produced by hot-air-roasted almonds. In addition, the moisture content of the almonds roasted by RF energy was lower, causing a crispier texture that scored 6.77 points, which was significantly higher than the 6.03 points garnered by the commercial almonds, while the other five sensory attributes eared scores greater than 6 points. The overall acceptance score of the RF-roasted almonds was 6.58 points higher than that of the commercial almonds (6.15 points).

## 4. Conclusions

An increased loading amount and decreased electrode plate gap caused a higher RF output power and faster heating rate. Under the treatment conditions of 1 kg almonds with a 10 cm distance between the RF electrode plates, it only took 3.5 min to reach the target roasting temperature of 120 °C, while traditional oven roasting required 120 min. In addition, the moisture content, water activity, acid value, and DPPH radical scavenging ability of the RF-roasted almonds were better than those of the commercial products. The results of the GC–IMS aroma analysis showed that the almonds produced more aromatic components after roasting, and the aroma of the RF-roasted almonds was different from that of the commercial almonds according to PCA analysis. The overall score of the RF-roasted almonds was higher for the sensory evaluation. Therefore, the RF roasting of almonds is a time-saving, effective, and quality-enhancing technique.

## Figures and Tables

**Figure 1 foods-11-01885-f001:**
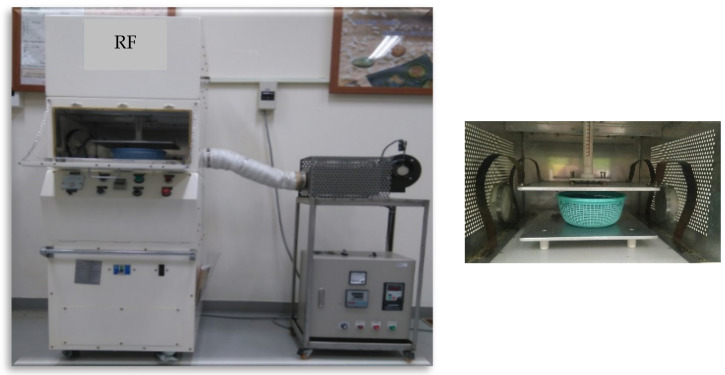
5 kW, 40.68 MHz RF hot air equipment (**left**) and a sample basket placed between RF electrodes (**right**).

**Figure 2 foods-11-01885-f002:**
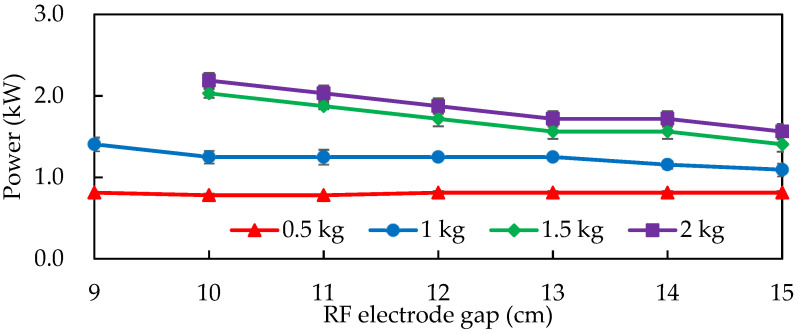
Effect of different loading levels of almonds and different 5 kW, 40.68 MHz RF electrode plate gaps on power. Data are expressed as mean ± S.D. (*n* = 3).

**Figure 3 foods-11-01885-f003:**
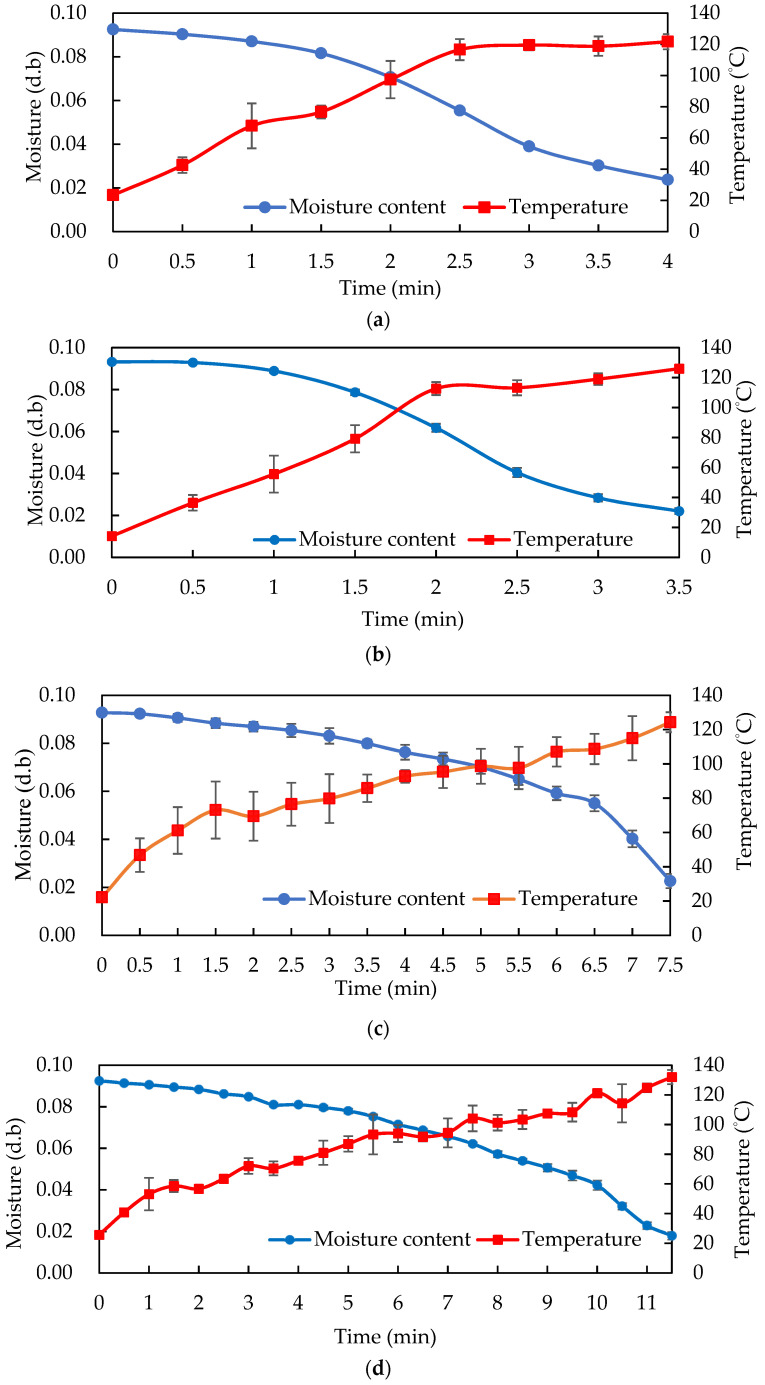
The drying and temperature curves of (**a**) 0.5, (**b**) 1, (**c**) 1.5, and (**d**) 2 kg almonds during 5 kW, 40.68 MHz batch RF roasting at 10 cm electrode gap. Data are expressed as mean ± S.D. (*n* = 3).

**Figure 4 foods-11-01885-f004:**
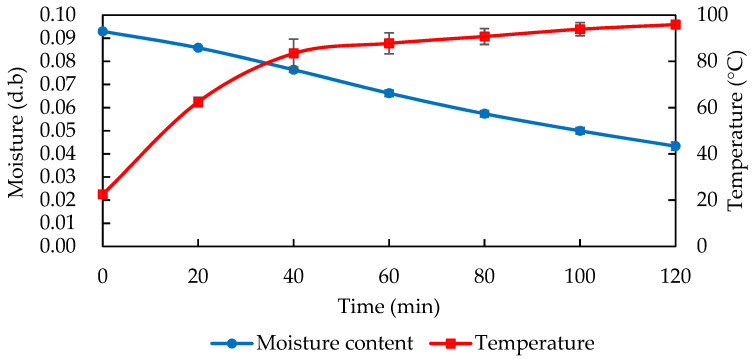
The drying curve and temperature profile of 1 kg almonds during 105 °C oven roasting. Data are expressed as mean ± S.D. (*n* = 3).

**Figure 5 foods-11-01885-f005:**
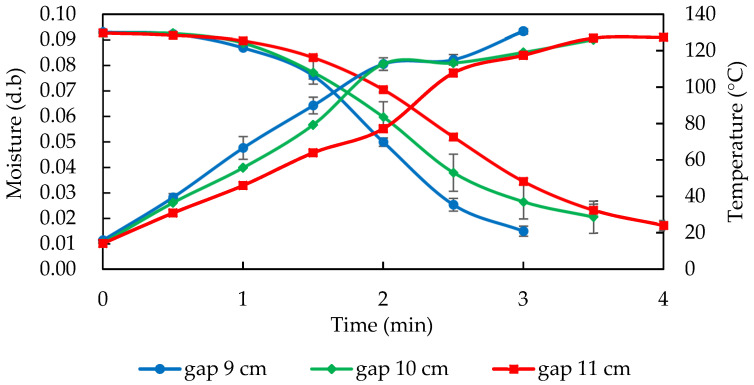
The drying and temperature curves of 1 kg almonds during 5 kW, 40.68 MHz RF roasting at different electrode gaps. Data are expressed as mean ± S.D. (*n* = 3).

**Figure 6 foods-11-01885-f006:**
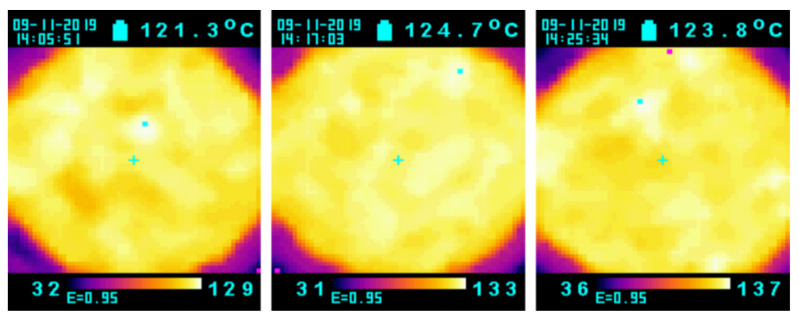
The surface temperature of 1 kg almonds after 5 kW, 40.68 MHz RF roasting (**left**: 9 cm gap, 3 min; **center**: 10 cm gap, 3.5 min; and **right**: 11 cm gap, 4 min).

**Figure 7 foods-11-01885-f007:**
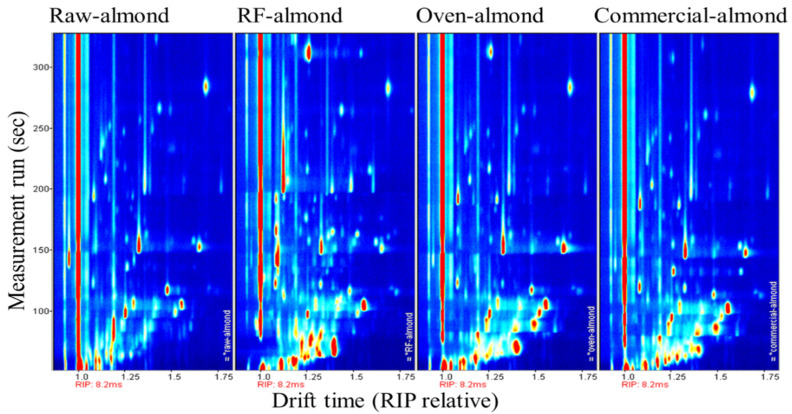
GC–IMS plot comparison of raw, RF-roasted, oven-roasted, and commercial almonds.

**Figure 8 foods-11-01885-f008:**
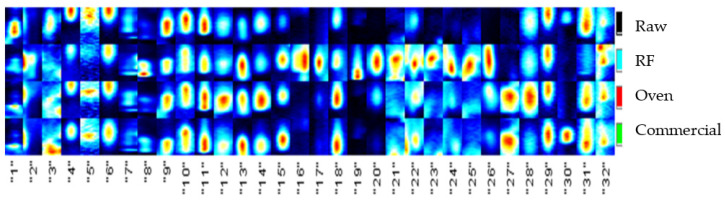
The characteristic aroma fingerprint of raw, RF-roasted, oven-roasted, and commercial almonds.

**Figure 9 foods-11-01885-f009:**
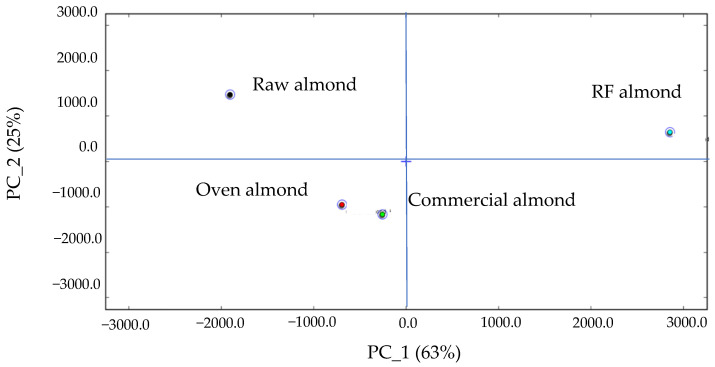
Principal component analysis of raw, RF-roasted, oven-roasted, and commercial almonds.

**Table 1 foods-11-01885-t001:** Quality parameters of raw, RF-roasted, conventional-oven-roasted, and commercial almonds.

Sample	Raw	RF	Oven	Commercial
MC (%)	8.47 ± 0.12 ^a^	1.57 ± 0.06 ^d^	3.6 ± 0.10 ^b^	2.3 ± 0.26 ^c^
Aw	0.74 ± 0.00 ^a^	0.34 ± 0.03 ^c^	0.49 ± 0.01 ^b^	0.56 ± 0.01 ^b^
L*	44.43 ± 0.72 ^a^	41.72 ± 0.84 ^b^	41.10 ± 0.84 ^b^	41.64 ± 0.59 ^b^
a*	15.58 ± 0.17 ^a^	15.56 ± 0.11 ^a^	14.32 ± 26.10 ^c^	15.08 ± 0.22 ^b^
b*	29.36 ± 0.26 ^a^	26.70 ± 0.54 ^b^	26.10 ± 0.42 ^c^	26.67 ± 0.51 ^b^
AV (mg/g)	0.58 ± 0.09 ^a^	0.34 ± 0.02 ^c^	0.48 ± 0.06 ^b^	0.47 ± 0.02 ^b^
DPPH (%)	82.50 ± 0.62 ^a^	64.60 ± 0.46 ^c^	72.20 ± 0.36 ^b^	52.10 ± 1.01 ^d^

Data are expressed as mean ± S.D (*n* = 5). Means with different superscript letters in the same row are significantly different (*p* < 0.05).

**Table 2 foods-11-01885-t002:** Consumer 9-point-scale hedonic sensory evaluation of RF-roasted and commercial roasted almonds.

Roasting	Appearance	Aroma	Flavor	Texture	Aftertaste	Overall
RF	6.57 ± 1.33	6.35 ± 1.78	6.63 ± 1.57 *	6.77 ± 1.49 *	6.75 ± 1.79	6.85 ± 1.58 *
Commercial	6.71 ± 1.28	5.85 ± 1.55	6.03 ± 1.58	6.03 ± 1.85	6.20 ± 1.66	6.15 ± 1.55

Data are expressed as mean ± S.D. (*n* = 65). Means with * in each column are significantly different (*p* < 0.05).

## Data Availability

The data presented in this study are available in this article.

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
