# Peer review of "Developing Radio-Frequency Roasting Protocols for Almonds Based on Quality Evaluations"

_foods, 2022, doi:10.3390/foods11131885_

Round 1

Reviewer 1 Report

Title: The title did not provide a focus of the study, suggest elaborate.

Line 38: As to my understanding, the commonly used roasting condition is much higher than 100 C, please check.

Line 165: the section title has some problem, please check.

Figure 1: the error bar (standard deviation) could be barely observed. Please provide.

Table 1. Would you like explaining why the time consumption of 0.5 kg sample was longer than that of 1 kg?

Figure 2: is the moisture content measurement repeated? The error bars can not be seen in the figures.

Table 2: ‘commercial’ was not supposed to be a treatment

Figure 7: the font size and shape in the figure were deformed, please revise

Also, I suggest more literature to be cited to make a more comprehensive literature review and to support the results in this study.

Author Response

Thank you for your comment and suggestions, we will answer each one.

Comments and Suggestions for Authors

Title: The title did not provide a focus of the study, suggest elaborate.

Answer: The title is changed to “Developing Radio Frequency Roasting Protocols for Almonds Based on Quality Evaluations”

Line 38: As to my understanding, the commonly used roasting condition is much higher than 100 C, please check.

Answer: the roasting method is higher than 100°C hot air heating for 120 minutes

Line 165: the section title has some problem, please check.

Answer: 2.4.6. GC-IMS analysis

Figure 1: the error bar (standard deviation) could be barely observed. Please provide.

Answer: We changed it in Figure 2.

Table 1. Would you like explaining why the time consumption of 0.5 kg sample was longer than that of 1 kg?

Answer: Yes. Because the RF power of 0.5 kg sample was much lower than that of 1 kg sample at same 10 cm gaps of the electrode plates during RF roasting.

Figure 2: is the moisture content measurement repeated? The error bars can not be seen in the figures.

Answer: We changed it in Figure 3.

Table 2: ‘commercial’ was not supposed to be a treatment

Answer: I changed the first column “treatment” to “sample”.

Figure 7: the font size and shape in the figure were deformed, please revise

Answer: The size and shape in the figure were changed to small size.

Also, I suggest more literature to be cited to make a more comprehensive literature review and to support the results in this study.

Reviewer 2 Report

In this work, authors investigated the influence of radio frequency on quality of almonds. The obtained results are interesting and presented in a clear way.

My recommendation are as follows:

I recommend rewriting and shortening the abstract by excluding some of the results, since it contains a lot of quantitative data without clearly emphasized aim of the study. Therefore, I recommend starting with the brief aim and then include methods, main results, and findings.

Please use formal writing style, instead of “they're” (Row 34) write full form.

“GC-IMS ssssss [11]” I recommend including the reference into the text, and not in the title of the section, please revise.

Please include full and detailed explanation of how the extract for DPPH test was prepared.

Author Response

Thank you for your comment and suggestions, we will answer each one.

Comments and Suggestions for Authors

 On my opinion the abstract is too long

Answer: The title is changed to “Developing Radio Frequency Roasting Protocols for Almonds Based on Quality Evaluations”

We delete “To begin, the power variation of 0.5, 1, 1.5, and 2 kg almonds was tested at various RF electrode plate gaps, and the result showed that the higher the loading capacity and the smaller the RF electrode plate gap, the higher the RF output power.” from the original abstract.

Ln 71-73. The objective should be much better described

Answer:  it is changed to “The objectives of this study were to develop a RF roasting protocol, to compare the quality of roasted almonds among commercial, oven and RF heating methods and to evaluate the aroma and sensory quality of RF roasted almonds.”

The section on material and methods is a bit chaotic an requires a lot of improvement

Answer: The section on materials and methods was divided to 2.1. Materials, 2.2. Equipment, 2.3. Roasting methods (2.3.1. Determination power output of RF roasting almonds, 2.3.2. The changes of temperature and moisture content of RF-roasting almonds, 2.3.3. Establishment of RF roasting almonds condition, 2.3.4. Conventional oven roasting almonds), 2.4. Analytical methods (2.4.1. Moisture content, 2.4.2. Water activity, 2.4.3. Color measurement, 2.4.4. DPPH radical scavenging ability assays, 2.4.5. Acid value, 2.4.6. GC-IMS analysis, 2.4.7. Sensory evaluation), and 2. 5. Statistical analysis.

In section 2.1-2.2 the purpose of some of the materials and equipment used should be much better described in the corresponding section. Maybe you can add the details on the equipment used when you described that the temperature was measured with an infrared temperature sensor. I think the origin of some of the chemical compounds used in this study should not be described in section 2.1, they are later described. It gives too many useless details.

Answer: The equipment was shown in Figure 1. 5 kW, 40.68 MHz RF hot air equipment (left) and a sample basket placed between RF electrodes (right).

The surface temperature was measured at the center and 5 cm on either side of the center with an infrared temperature sensor.

In Section 2.3.1 add a figure describing the setup.

Answer: We add Figure 1 for RF equipment and a sample basket.

Why a basket with holes?

Answer: The basket with holes will help hot air blowing to remove the evaporated water vapor.

In line 107 please describe the formula you are using

Answer: Because RF equipment has a maximum current of 1.6 Amp and a maximum output power of 5 kW, the average output power of the RF system is calculated by reading current (A), and using the formula power output (kW) = (5/1.6) A.

Ln 114-116 another example that the methods are not correctly described. This could be replaced with a citation. I think this is already described in the analytical methods.

Answer:  The dry base moisture content change was measured by the weight change of the sample during the RF roasting. We delete “which required the use of 105°C oven to dry the sample to constant weight in order to determine the dry rate of the sample.

Moisture content (d.b.) = (Wt-Wo)/Wo

Where Wt is the weight of sample with drying time t, Wo is the initial weight of sample × dried solid content.”

Most of the analytical methods could be replaced by a citation. This paper has too few citations.

Answer: We had citations in DPPH assay [9], The acid value was determined according to Chinese National Standard analytical methods for edible oil (CNS 3647 N6082) [10], GC-IMS analysis [11].

I think the results section should be reorganized, it is a bit chaotic and difficult to follow. For instance, Fig1 and Fig3 should be discussed together o Fig2 and Fig5. There are some comments that can be read in different sections.

Answer: We want to select a suitable gap from Figure 2. And then we choose at gap of 10 cm for RF roasting different loading almonds in Figure 3. Because 1kg almonds had the fastest heating rate, then we operated at three different electrode gaps in Figure 4.  Finally we compared the drying curve and temperature profile of 1 kg almonds during 105°C oven roasting in Figure 6.

Section 3.1 how can be explained that the power does not change for 0.5 kg? I think that the results in section 3.1 is quite obvious.

Answer: Because the height of basket was 9.5 cm for 0.5 kg almonds, and the height of sample was too small for changing the electrode gaps (9cm to 15 cm).

Table 1. I do not understand the linear regression equation. Is it necessary?

Answer: The linear regression equation from average temperature change during RF roasting in Figure 2 (A), (B), (C), and (A)), and the increasing temperature rate could be evaluated in different loading. 

Very few related studies are found in the discussion. This should be improved.

Answer: There are currently few studies on the application of radio frequency roasting nuts, only in peanuts, cashews, and almonds.

We add one reference Tu, X. H., Wu, B. F., Xie, Y., Xu, S. L., Wu, Z. Y., Lv, X., Wei, W., Du, L., & Chen, H. (2021). A comprehensive study of raw and roasted macadamia nuts: Lipid profile, physicochemical, nutritional, and sensory properties. Food Science & Nutrition, 9(3), 1688-1697. “The sensory, nutritional quality, and oxidative stability of roasted macadamia nuts were greatly improved, compared with raw nuts.”

Table 3. Why do you compare RF roasting to commercial and to oven roasting?? This does not make sense to me.

Answer: The commercial almond was roasted by hot air oven. We use 105℃ oven to roast almonds. We want to compare their quality to RF roasted almonds.

Conclusions is not a summary of the study. Implications of this study should be pointed out.

Answer: Under the treatment of 1 kg almond with 10 cm distance between RF electrode plates, it only took 3.5 min to reach the target roasting temperature of 120°C, while traditional oven roasting required 120 min. RF roasting almonds are a time-saving, effective and quality-enhancing technique.

Reviewer 3 Report

The authors need to say where they sourced the commercial roasted almonds and whether they were from the same original supplier as the raw almonds used in the rest of the study. Could the differences in the characteristics of commercially roasted almonds compared to the RF and Oven roasted almonds be due to differences in the variety, source, size, or age of the products?

The subtitles on lines 120 and 165 need to be corrected.

The plots in Figure 2 are misleading because the x axes are different lengths. It would be better to plot all the temperature curves in Fig 2(a) and all the moisture levels in Fig 2(b).

Line 290 suggests the oven temperature was 105 degrees but elsewhere it is 120 degrees. Which is correct?

I like the approach of GC-IMS but I believe the results plotted in Figure 6 are especially in Figure 7 are incomplete. How can you call 30 unidentified spots a signature? You should be able to identify some or all of these compounds by MS library searches and MW and fragment characterisation. It sould be possible to identify many of the 30 compounds to make this section complete.

Author Response

Thank you for your comment and suggestions, we will answer each one.

Comments and Suggestions for Authors

The authors need to say where they sourced the commercial roasted almonds and whether they were from the same original supplier as the raw almonds used in the rest of the study. Could the differences in the characteristics of commercially roasted almonds compared to the RF and Oven roasted almonds be due to differences in the variety, source, size, or age of the products?

Answer: The almonds used in this study were purchased from Beans Group Foods Science and Technology Co. and the origin was California, USA. We bought raw almonds and commercial almonds from the same company (Beans Group Foods Science and Technology Co., LTD.).

The subtitles on lines 120 and 165 need to be corrected.

Answer: We change Line 120 “RF roasting almonds condition” and Line 165 ” GC-IMS analysis”.

The plots in Figure 2 are misleading because the x axes are different lengths. It would be better to plot all the temperature curves in Fig 2(a) and all the moisture levels in Fig 2(b).

Answer: We want that graphics should be consistent; therefore, they are presented in their current state.

Line 290 suggests the oven temperature was 105 degrees but elsewhere it is 120 degrees. Which is correct?

Answer: We used 105℃ oven to roast in this study.

I like the approach of GC-IMS but I believe the results plotted in Figure 6 are especially in Figure 7 are incomplete. How can you call 30 unidentified spots a signature? You should be able to identify some or all of these compounds by MS library searches and MW and fragment characterisation. It sould be possible to identify many of the 30 compounds to make this section complete.

Answer: We can’t find the GC-IMS library to identify the 30 compounds. However, it was very clear to distinguish aroma difference in almonds by different roasting processing.  

Reviewer 4 Report

On my opinion the abstract is too long

Ln 71-73. The objective should be much better described

The section on material and methods is a bit caotic an requires a lot of improvement

In section 2.1-2.2 the purpose of some of the materials and equipment used should be much better described in the corresponding section.. Maybe you can add the details on the equipment used when you described that the temperature was measured with a infrared temperature sensor. I think the origin of some of the chemical compounds used in this study should not be described in section 2.1, they are later described. It gives too many useless details.

In Section 2.3.1 add a figure describing the setup.

Why a basket with holes?

In line 107 please describe the formula you are using

Ln 114-116 another example that the methods are not correctly described. This could be replaced with a citation. I think this is already described in the analytical methods.

Most of the analytical methods could be replaced by a citation. This paper has too few citations.

I think the results section should be reorganized, it is a bit caotic and difficult to follow. For instance, Fig1 and Fig3 should be discussed together o Fig2 and Fig5. There are some comments that can be read in different sections.

Section 3.1 how can be explained that the power does not change for 0.5 kg? I think that the results in section 3.1 is quite obvious.

Table 1. I do not understand the linear regression equation. Is it necessary?

Very few related studies are found in the discussion. This should be improved.

Table 3. Why do you compare RF rosting to commercial and to oven roasting?? This does not make sense to me.

Conclussions is not a summary of the study. Implications of this study should be pointed out.

Author Response

The reviewer 4 are the same as the reviewer 2. 

Thank you for your comment and suggestions, we will answer each one.

Comments and Suggestions for Authors

 On my opinion the abstract is too long

Answer: The title is changed to “Developing Radio Frequency Roasting Protocols for Almonds Based on Quality Evaluations”

We delete “To begin, the power variation of 0.5, 1, 1.5, and 2 kg almonds was tested at various RF electrode plate gaps, and the result showed that the higher the loading capacity and the smaller the RF electrode plate gap, the higher the RF output power.” from the original abstract.

Ln 71-73. The objective should be much better described

Answer:  it is changed to “The objectives of this study were to develop a RF roasting protocol, to compare the quality of roasted almonds among commercial, oven and RF heating methods and to evaluate the aroma and sensory quality of RF roasted almonds.”

The section on material and methods is a bit chaotic an requires a lot of improvement

Answer: The section on materials and methods was divided to 2.1. Materials, 2.2. Equipment, 2.3. Roasting methods (2.3.1. Determination power output of RF roasting almonds, 2.3.2. The changes of temperature and moisture content of RF-roasting almonds, 2.3.3. Establishment of RF roasting almonds condition, 2.3.4. Conventional oven roasting almonds), 2.4. Analytical methods (2.4.1. Moisture content, 2.4.2. Water activity, 2.4.3. Color measurement, 2.4.4. DPPH radical scavenging ability assays, 2.4.5. Acid value, 2.4.6. GC-IMS analysis, 2.4.7. Sensory evaluation), and 2. 5. Statistical analysis.

In section 2.1-2.2 the purpose of some of the materials and equipment used should be much better described in the corresponding section. Maybe you can add the details on the equipment used when you described that the temperature was measured with an infrared temperature sensor. I think the origin of some of the chemical compounds used in this study should not be described in section 2.1, they are later described. It gives too many useless details.

Answer: The equipment was shown in Figure 1. 5 kW, 40.68 MHz RF hot air equipment (left) and a sample basket placed between RF electrodes (right).

The surface temperature was measured at the center and 5 cm on either side of the center with an infrared temperature sensor.

In Section 2.3.1 add a figure describing the setup.

Answer: We add Figure 1 for RF equipment and a sample basket.

Why a basket with holes?

Answer: The basket with holes will help hot air blowing to remove the evaporated water vapor.

In line 107 please describe the formula you are using

Answer: Because RF equipment has a maximum current of 1.6 Amp and a maximum output power of 5 kW, the average output power of the RF system is calculated by reading current (A), and using the formula power output (kW) = (5/1.6) A.

Ln 114-116 another example that the methods are not correctly described. This could be replaced with a citation. I think this is already described in the analytical methods.

Answer:  The dry base moisture content change was measured by the weight change of the sample during the RF roasting. We delete “which required the use of 105°C oven to dry the sample to constant weight in order to determine the dry rate of the sample.

Moisture content (d.b.) = (Wt-Wo)/Wo

Where Wt is the weight of sample with drying time t, Wo is the initial weight of sample × dried solid content.”

Most of the analytical methods could be replaced by a citation. This paper has too few citations.

Answer: We had citations in DPPH assay [9], The acid value was determined according to Chinese National Standard analytical methods for edible oil (CNS 3647 N6082) [10], GC-IMS analysis [11].

I think the results section should be reorganized, it is a bit chaotic and difficult to follow. For instance, Fig1 and Fig3 should be discussed together o Fig2 and Fig5. There are some comments that can be read in different sections.

Answer: We want to select a suitable gap from Figure 2. And then we choose at gap of 10 cm for RF roasting different loading almonds in Figure 3. Because 1kg almonds had the fastest heating rate, then we operated at three different electrode gaps in Figure 4.  Finally we compared the drying curve and temperature profile of 1 kg almonds during 105°C oven roasting in Figure 6.

Section 3.1 how can be explained that the power does not change for 0.5 kg? I think that the results in section 3.1 is quite obvious.

Answer: Because the height of basket was 9.5 cm for 0.5 kg almonds, and the height of sample was too small for changing the electrode gaps (9cm to 15 cm).

Table 1. I do not understand the linear regression equation. Is it necessary?

Answer: The linear regression equation from average temperature change during RF roasting in Figure 2 (A), (B), (C), and (A)), and the increasing temperature rate could be evaluated in different loading. 

Very few related studies are found in the discussion. This should be improved.

Answer: There are currently few studies on the application of radio frequency roasting nuts, only in peanuts, cashews, and almonds.

We add one reference Tu, X. H., Wu, B. F., Xie, Y., Xu, S. L., Wu, Z. Y., Lv, X., Wei, W., Du, L., & Chen, H. (2021). A comprehensive study of raw and roasted macadamia nuts: Lipid profile, physicochemical, nutritional, and sensory properties. Food Science & Nutrition, 9(3), 1688-1697. “The sensory, nutritional quality, and oxidative stability of roasted macadamia nuts were greatly improved, compared with raw nuts.”

Table 3. Why do you compare RF roasting to commercial and to oven roasting?? This does not make sense to me.

Answer: The commercial almond was roasted by hot air oven. We use 105℃ oven to roast almonds. We want to compare their quality to RF roasted almonds.

Conclusions is not a summary of the study. Implications of this study should be pointed out.

Answer: Under the treatment of 1 kg almond with 10 cm distance between RF electrode plates, it only took 3.5 min to reach the target roasting temperature of 120°C, while traditional oven roasting required 120 min. RF roasting almonds are a time-saving, effective and quality-enhancing technique.

Round 2

Reviewer 3 Report

Line 11. RF is used but has not yet been given in full.

Lines 24-25. There is repetition in these two sentences.

Lines 25-26. The conclusion suggests the RF roasted almonds had better appearance, aroma and aftertaste than commercial almonds but Table 9 shows there is no significant difference between the groups. So this statement is incorrect.

Line 94 does not make sense. What is suggested by "drying pesticide"? Why should pesticide be required if heating causes disinfestion as suggested?

Line 239. Does not make sense. "IMS ionization zone with a radiation The ion source with a radiation"?

Line 532. It suggests that the aroma compounds 16-26 are stronger than the RF treated samples but it looks to me like the RF treated samples have the highest intensity signals. I believe "than" is not the correct word. Also, why are compounds 27-32 less desirable than compounds 16-26?

Lines 533 and 539. The authors indicate that the commercial sample was roasted at 100 degrees C but how do they know this? Did they contact the roasting company? Or is it speculation?

I do not understand why the identity of compounds separated by GC cannot be compared to existing GC-MS data libraries. GC data without presumed identities of compounds is difficult to publish.

Author Response

Thank your comments and we answer one by one. Because round two reviewers gave us many comments, the manuscript was revised in many places. Pleased review the attached revise file.

Comments and Suggestions for Authors

Line 11. RF is used but has not yet been given in full.

Answer: We change “batch radio frequency (RF) equipment”

Lines 24-25. There is repetition in these two sentences.

Answer: We change it.

Lines 25-26. The conclusion suggests the RF roasted almonds had better appearance, aroma and aftertaste than commercial almonds but Table 9 shows there is no significant difference between the groups. So this statement is incorrect.

Answer: We change to “The RF roasted almonds had a better flavor, texture, and overall preference than commercial almonds.”

Line 94 does not make sense. What is suggested by "drying pesticide"? Why should pesticide be required if heating causes disinfestion as suggested?

Answer: We delete “pesticide”.

Line 239. Does not make sense. "IMS ionization zone with a radiation The ion source with a radiation"?

Answer: We change to “and then the gas was separated into the IMS ionization zone chamber.”

Line 532. It suggests that the aroma compounds 16-26 are stronger than the RF treated samples but it looks to me like the RF treated samples have the highest intensity signals. I believe "than" is not the correct word. Also, why are compounds 27-32 less desirable than compounds 16-26?

Answer: We change to “while in 16-26, the flavor signal was the highest intensity signals in the roasted almonds at 120°C RF heating.”

Lines 533 and 539. The authors indicate that the commercial sample was roasted at 100 degrees C but how do they know this? Did they contact the roasting company? Or is it speculation?

Answer: We purchased raw almonds and commercial almonds from Beans Group Foods Science and Technology Co. and contacted them to inquire about their roasting almonds processing.

I do not understand why the identity of compounds separated by GC cannot be compared to existing GC-MS data libraries. GC data without presumed identities of compounds is difficult to publish.

Answer: We did not use GC-MS to separate aroma of almonds, however, GC-IMS is a convenient and good instrument for distinguish the aroma compound by different processing or formula.

Reviewer 4 Report

Abstract.

Please indicate the meaning of RF before using it in the text.

I still find the abstract too long.

I also miss a sentence indicating the summarizing the objective of this study.

Material and Methods

This section is for me still quite chaotic. Sometimes processing methods are mixed up with analytical methods. I think still needs a lot of improvement.

I do not see the point of sections 2.1 (Materials) and 2.2 (Equipments). This should be included when the corresponding method is decribed.

Section 2.3.1 Determination power output of RF roasting almonds 

Is this the rigth title?? I do not see the relation between the title and the content.

Section 2.3.2. , 2.3.3

I think a better title should be found.

Section 2.4.1

I think the formula is not necessary

Section 2.4.4-2.4.6

If there is an official method, please cite it. There is no need for describing methods that are much better described in the literature.

Table 1.

I do not see the need for this equations.

Line 227-228

"Compared with the conventional hot air technology, the RF roasting almonds showed a fast and stable temperature increase without any significant heat transfer resistance."

Where are the results or the citation?

Sensory analysis:

Does the different water content of commercial and RF almonds affect the sensory evaluation?

Table. Why did you not include the results of oven processed almonds?

Conclusion should be improved. The implications of the results should be detailed.

Author Response

Thank your comments and we answer one by one. Because round two reviewers gave us many comments, the manuscript was revised in many places. Pleased review the attached revise file.

Comments and Suggestions for Authors

Abstract.

Please indicate the meaning of RF before using it in the text.

I still find the abstract too long.

I also miss a sentence indicating the summarizing the objective of this study.

Answer: The abstract is changed to “Hot air roasting is a popular method for almonds, but it takes a long time. The almonds were roasted using dielectric heating with 5 kW, 40.68 MHz batch radio frequency (RF) equipment, and their quality and aroma were analyzed using a gas chromatography-ion shift spectrometer and sensory evaluation. Almonds with an initial moisture content of 8.47 % were heated at RF electrode gap of 10 cm, and a target roasting temperature of 120°C was achieved at different weights of 0.5, 1, 1.5, and 2 kg for 4, 3.5, 7.5, and 11 min, respectively, and the moisture content was reduced to less than 2%. In comparison, 1 kg of almonds roasted in a 105°C conventional oven for 120 min. The darker color, lower moisture content, water activity, and acid value of RF roasted almonds were favorable for preservation. Aroma analysis using a gas chromatography- ion mobility spectroscopy (GC-IMS) revealed that the aroma signal after roasting was richer than that of raw almonds, and principal component analysis (PCA) demonstrated that the aroma of roasted and commercial almonds was similar. The RF roasted almonds had a better flavor, texture, and overall preference than commercial almonds. Radio frequency heating will be used in the food industry to roast nuts.”.

Material and Methods

This section is for me still quite chaotic. Sometimes processing methods are mixed up with analytical methods. I think still needs a lot of improvement.

 I do not see the point of sections 2.1 (Materials) and 2.2 (Equipments). This should be included when the corresponding method is described.

Answer: We delete 2.2 and add the equipment information in the following method description.

Section 2.3.1 Determination power output of RF roasting almonds 

Is this the right title?? I do not see the relation between the title and the content.

Section 2.3.2. , 2.3.3

I think a better title should be found.

Answer: We delete 2.3.2 and 2.3.3 and change to “2.2.1. Determination of RF roasting almonds condition

Section 2.4.1

I think the formula is not necessary

Answer: We delete the formula.

Section 2.4.4-2.4.6

If there is an official method, please cite it. There is no need for describing methods that are much better described in the literature.

Answer: We delete some description in section 2.3.4-2.3.5, but we just delete some description in section 2.3.6 the GC-IMS aroma analysis.

Table 1.

I do not see the need for this equations.

Answer: We delete Table 1.

Line 227-228

"Compared with the conventional hot air technology, the RF roasting almonds showed a fast and stable temperature increase without any significant heat transfer resistance."

Where are the results or the citation?

Answer: We delete it and move the original Figure 6 to Figure 4, then they can be compared.

Sensory analysis:

Does the different water content of commercial and RF almonds affect the sensory evaluation?

Answer: It may cause difference of the sensory evaluation.

Table. Why did you not include the results of oven processed almonds?

Answer: The sample size of oven roasted almonds was not enough for sensory evaluation.

Conclusion should be improved. The implications of the results should be detailed.

Answer: In the last sentence, we have mentioned “Therefore, RF roasting almonds are a time-saving, effective and quality-enhancing technique.”
